# A Novel Approach for Improving XML Querying over Wireless Broadcast Channels

**DOI:** 10.3390/s24227206

**Published:** 2024-11-11

**Authors:** Vinay Kumar Ahlawat, Gaurav Agarwal, Vikas Goel, Akash Sanghi, Sun Young Choi, Kueh Lee Hui, Mangal Sain

**Affiliations:** 1Department of CSE, Invertis University, Bareilly 243123, India; vinahlawat@gmail.com (V.K.A.); gaurav.a1@invertis.org (G.A.); akash.s@invertis.org (A.S.); 2Department of IT, KIET Group of Institutions, Ghaziabad 201206, India; rvikasgoel@yahoo.com; 3Department of Electrical Engineering, Dong-A University, Busan 49236, Republic of Korea; drputt@dau.ac.kr; 4Division of Computer & Information Engineering, Busan 49236, Republic of Korea

**Keywords:** XML, querying, wireless broadcast channels, partitioning, load balancing, query routing, result aggregation, caching, access time

## Abstract

The querying of large XML data over wireless broadcast channels can reduce bandwidth utilization, cause significant latency, and produce inefficient energy usage. This paper proposes a scheme to improve XML querying over wireless broadcast channels in order to address the issues mentioned above. Various techniques, including partitioning, load balancing, and query routing, have been combined into one approach. The proposed scheme partitions the XML data stream into several partitions based on criteria like document size, type, or content. Each partition is routed to a separate channel to balance the load on each wireless broadcast channel. A query routing mechanism that directs queries to the right channel or combination of channels that hold the relevant XML data partition was implemented. This study simulates, evaluates, and compares the proposed scheme’s performance. The results from the comparison study with existing schemes demonstrate a considerable reduction in the access time for XML querying via wireless broadcast channels.

## 1. Introduction

Wireless broadcast channels are becoming increasingly popular for delivering large amounts of XML data to mobile devices in a variety of wireless network environments. Broadcasting data over a wireless channel adds index information that indicates when the data arrived. This is an energy-efficient option [1]. When the necessary data are available on the channel, indexing adds arrival information to it. The broadcast environment (single or multiple channels), data type (simple or XML), and broadcast schedule all have an impact on indexing approaches [2]. A single-channel environment only sends data items and indexes over one channel, as opposed to a multichannel environment, which transmits data across multiple channels. It is possible for a group of data items to be the same size or modifiable depending on the data type. The distribution of data across broadcast channels is determined by broadcast scheduling [3,4,5].

The latency and bandwidth constraints associated with wireless broadcast channels make accessing XML data difficult.

To address these issues, we propose a new XML data placement scheme that boosts the performance of XML querying over wireless broadcast channels. The proposed scheme partitions the XML data stream and distributes it across multiple channels, increasing parallelism and reducing access time during XML querying. Furthermore, the proposed scheme aids in balancing the load across different channels and improves fault tolerance and robustness in the face of network failures. This proposed work also discusses how the proposed scheme will be implemented, including partitioning criteria, query routing mechanisms, and challenges. Overall, the proposed XML data placement scheme shows promise for disseminating XML data via wireless broadcast channels, particularly in large-scale and data-intensive applications.

This proposed scheme has the potential to improve energy efficiency in mobile devices by reducing the time and energy required for XML querying. Mobile devices can perform the required queries more efficiently by accessing multiple partitions of the XML data stream at the same time, reducing the amount of time and energy required to transmit and receive data via wireless broadcast channels.

This scheme is widely applicable to a variety of applications involving large-scale, data-intensive XML data, including social networks, e-commerce, and healthcare. For example, in social networks, large amounts of XML data are generated and distributed among users, and the proposed scheme can help improve the performance and efficiency of XML querying in such systems. In e-commerce, this scheme can be used to distribute large product catalogs via wireless broadcast channels, allowing customers to search and browse more efficiently. In healthcare, this scheme can be used to disseminate patient data and medical records via wireless broadcast channels, allowing healthcare professionals to access medical information more quickly and efficiently [6,7,8].

The rest of the paper is organized as follows: Section 2 presents related work, Section 3 describes the proposed scheme in detail, Section 4 presents the simulation methodology and results, Section 5 discusses the performance evaluation and comparison with existing schemes, and Section 6 concludes the paper and suggests future work.

## 2. Related Work

The authors introduced a wireless XML streaming method that provides energy-efficient access to a wireless stream. To represent XML data and index information, the authors created two hierarchical structures: the XML data tree and the XML index tree. The wireless XML stream is created by traversing these two structures with specific duplications [9].

This study presented a novel XML stream structure for distributing XML data over a broadcast channel. The structure consisted of grouping and summarizing the structural information of XML nodes. It is possible to reduce the size of the XML stream by summarizing the information, thus resulting in a faster retrieval of the desired XML data via a wireless broadcast channel. The suggested XML stream structure includes indexes that allow you to skip unimportant parts of the XML stream. As a result, it can reduce the energy consumption of mobile devices when receiving XML query results [10].

Path Stream Group Level (PSGL) nodes are introduced as new frameworks for processing streaming XML data. The framework takes advantage of the hierarchical structure of XML documents. Mobile clients can selectively download XML data using various tiny indices, such as level, child, sibling, attribute, and text. The data were organized using the XML document tree’s hierarchy, and XML items with the same XML path prefix were grouped to save battery life on mobile clients [11].

This paper described two novel XML data placement strategies for distributing XML data over multiple wireless broadcast channels. The XML data stream was divided into several sections and assigned to various wireless broadcast channels. The goal of both was to reduce access time when running XML queries across broadcast channels. Based on the experimental results, the proposed XML data placement strategies achieved the goal of further reducing access time, especially when dealing with large XML documents [12].

Between the beginning of human civilization and the year 2000, humanity produced a total of five exabytes of data. The writers were, however, producing five exabytes of data every day. It loses its usefulness if no query operation is run on the data.

Any organization that analyses these data may become more resilient and nimbler, enabling it to handle a range of business issues.

The query-processing system is an important component of big data because it manages queries. This study distinguished between several query optimization procedures and their algorithms, which are used in large datasets to avoid query optimization problems. This review assisted researchers in identifying alternative methods for processing data and improving query processing and optimization in a variety of applications [13] by employing different query processing strategies.

This paper investigated the use of wireless network connectivity to store Wushu Historical Archives’ XML metadata database. The study focused on metadata storage in XML databases, specifically document size, document loading, and document query. This is achieved by using data load-balancing algorithms and two-way path constraint algorithms [14].

The authors of this research study proposed a hybrid indexing approach for data transmission. This method distributes the indexing using a hash table and employs Huffman-tree index coding. The authors of the proposed work conducted a thorough analysis to provide a theoretical explanation for the current indexing scheme’s performance. The outcomes of different strategies were then compared.

The authors used an indexing method to retrieve geographic information in a wire-free environment [15] in their proposed solution.

## 3. Proposed Work

A new XML data placement scheme is proposed to shorten access time during XML querying via wireless broadcast channels. The proposed scheme divides the XML data stream into several partitions and distributes them across multiple wireless broadcast channels. As a result, mobile clients can simultaneously access different partitions of the XML data stream from various broadcast channels, reducing access time even further.

The proposed XML querying & data placement scheme involves the following steps:Read the XML data and queryDivide the XML data stream into several partitions based on criteria, such as document size (small/large), document type (role/schema), or document content (key characteristics/metadata).Assign each partition to a different wireless broadcast channel, ensuring that the load on each channel is balanced.Broadcast the XML data partitions over their respective channels.Implement a query routing mechanism that can route the queries to the appropriate channel or combination of channels that contain the relevant XML data partition.The query result is forwarded to a mobile device

Figure 1 illustrates the steps involved in the proposed XML querying process over wireless broadcast channels.

This suggested method shortens access times during XML querying, especially for big XML data streams.

This is due to mobile clients concurrently accessing multiple partitions of the XML data stream via different broadcast channels. This increases parallelism and lowers the latency of XML queries. Our proposed XML data placement scheme has the potential to improve XML query performance on wireless broadcast channels.

The proposed scheme can be used in a variety of wireless broadcasting systems, including DAB, DVB, and ISDB-T [16]. The partitioning criteria can be determined by the characteristics of the XML data and the application’s requirements. For example, if the XML data stream contains a variety of documents of varying sizes, it may be beneficial to partition them based on document size to balance the load across each channel. On the other hand, if the XML data stream contains similar documents, they can be partitioned by document type or content to improve query performance.

Query routing can be implemented using a variety of techniques, such as query decomposition, query caching, and query broadcasting. During query decomposition, the query is divided into sub-queries, each of which is routed to the channel with the relevant XML data partition. Query caching stores queries and their results in the mobile device cache to prevent future redundant queries. Query broadcasting involves broadcasting the query to all channels and then combining the results to produce the final result.

The proposed scheme has several advantages, including improved query performance, shorter access times, and greater parallelism. The overhead of dividing and spreading the XML data stream, the requirement that mobile devices support numerous wireless broadcast channels, and the necessity for effective query routing and load balancing techniques are some of its drawbacks.

The suggested XML data insertion technique helps to balance the load across many wireless broadcast channels while also cutting down on access time.

This is because the XML data partitions are evenly distributed across the available channels, ensuring that each channel transmits a manageable amount of data. This can help to avoid overloading any single channel, which can result in poor performance and increased energy consumption on mobile devices.

Another benefit of this scheme is that it can improve fault tolerance and robustness in the event of network failures or disruptions. If one channel becomes unavailable or experiences a disruption, the XML data partitions can still be accessed through the remaining channels. This can help to ensure that mobile clients always have access to the necessary XML data, even in the event of a network failure or other disruption.

### 3.1. Architecture Diagram for Proposed XML Querying Scheme

Figure 2 depicts the proposed architecture for the XML querying scheme. The diagram depicts a straightforward method for partitioning an XML data stream based on a specified criterion. Any wireless device can submit a query to the server via a Query Interface. The query is then routed to the appropriate channel, allowing the XML data source to return the result. These results are gathered and combined using an XML query result aggregator. The final result is then sent to the specified wireless client. The steps ensure that the data are divided into partitions that reflect the distinct values of the chosen criterion, allowing for more efficient processing and querying.

The steps for the XML data partition and broadcasting in the architecture of the proposed XML querying scheme as per the data access algorithm are as follows:Reading the XML Data Stream:The system starts by reading the input XML data stream.Choose Partitioning Criterion:Select the criterion for partitioning, such as document size, document type, or document content.For Each Document in the XML Data Stream:Iterate through each document in the XML data stream.Determine the Value of the Chosen Criterion:For each document, determine the value of the chosen partitioning criterion.Create Partition for Each Unique Criterion Value:Create a separate partition for each unique value of the chosen criterion.Assign Document to Corresponding Partition:Assign the document to the partition that corresponds to its criterion value.Repeat Steps 3–6 for All Documents:Continue processing each document in the XML data stream according to the above steps.Output Partitions as Separate XML Data Streams:Output the resulting partitions as separate XML data streams.

To partition an XML data stream, first read the input XML data stream. The system begins by ingesting and preparing this stream for processing. Next, a partitioning criterion is chosen, which could be based on document size, type, or content. Once the criterion is determined, the system iterates through each document in the XML data stream. It assigns a value to each document based on the partitioning criterion. Based on these values, the system generates a separate partition for each unique criterion value. Each document is then assigned to the partition corresponding to its specific criterion value. This process is repeated for each document in the XML data stream. It ensures that each document is correctly assigned to the appropriate partition. Finally, the partitions are output as separate XML data streams, with each representing a subset of the original data based on the partitioning criterion used. This methodical approach ensures that the XML data are efficiently organized for future querying and processing.

The steps for the XML query routing mechanism in the architecture of the proposed XML querying scheme as per the data access algorithm are as follows:Divide the XML Data Stream:XML Data Source: The origin of the XML data stream.Partitioning Process: The data stream is divided into several partitions based on predefined criteria such as document size, type, or content.Assign Partitions to Wireless Broadcast Channels:Each partition is assigned to a different wireless broadcast channel, ensuring balanced load distribution across all channels.Broadcast the XML Data Partitions:Wireless Broadcast Channels: Each channel broadcasts its assigned XML data partition.Implement Query Routing Mechanism:Query Router: Directs user queries to the appropriate channels.Query Interface: Where users input their queries.Routing Logic: Routes the query to the relevant channel or channels based on the XML data partitions they contain.Result Collection and Combination: Collect results from each channel and combine them to generate the final query result.Result Delivery: Returns the combined query results to the user.

The proposed scheme for improving XML querying over wireless broadcast channels consists of several key steps. To begin, the XML data stream, which originates from an XML data source, is partitioned based on predefined criteria such as document size, type, and content. This partitioning process organizes the data for efficient querying. Next, each partition is assigned to a different wireless broadcast channel to ensure that the load is distributed evenly across all channels. Each wireless broadcast channel then transmits the assigned XML data partition. To handle user queries, a query routing mechanism is used. This includes a Query Router, which routes user queries to the appropriate channels, and a Query Interface, where users enter their queries. The routing logic directs queries to the appropriate channels based on the XML data partitions they contain. After the queries are processed, the system gathers the results from each channel and combines them to produce the final query result. After the query process is completed, the combined query results are returned to the user, ensuring that the query process is completed efficiently and effectively.

This architecture ensures the efficient and balanced querying of XML data over wireless broadcast channels by dividing the data, assigning it to channels, broadcasting, and implementing a strong query routing mechanism.

### 3.2. Detailed Description of the Architecture Components

The proposed architecture for improving XML querying over wireless broadcast channels involves several components working together seamlessly.
(i)Mobile Device: On the mobile device, users enter queries into a Query Interface, which then routes these queries to the appropriate channels or retrieves cached results if available. Furthermore, the Query Decomposer divides complex queries into sub-queries and communicates with the cache to check for existing results or replace them with new ones.Query Interface: Where users input their queries.Query Router: Routes queries to the appropriate channels or cached results.Query Decomposer/Cache: Decomposes complex queries into sub-queries and checks/updates the cache.(ii)XML Data Partitions: The XML data are divided into segments based on specific criteria such as document size, type, or content, and each partition is assigned to a separate wireless broadcast channel. This approach distributes the load and allows each channel to broadcast its assigned XML data segment efficiently.Document Size: XML documents of comparable sizes are grouped together by document size partitioning. For instance, one partition might include smaller documents, while another might contain larger documents. Because smaller files may be stored and retrieved faster, this method can effectively improve response times for simpler queries in networks where smaller files are visited more frequently. Larger papers, on the other hand, can be streamed independently to prevent using resources that are required for smaller, faster-access files.Document Type: Document type partitioning arranges the XML data according to the role or schema structure of the document. “Customer records,” “product catalogs,” and “transaction logs” are a few examples. Sorting by type eliminates the need to search through irrelevant data kinds by directing particular query types to the appropriate channel. When data have different access patterns for each type or when queries are type-specific, this method works very well.Document Content: Documents are divided using content-based partitioning according to specific important attributes or metadata found in the XML content itself. XML documents could be divided, for instance, by date, topic, or geographic location. This method makes it possible for queries aimed at particular content categories to be effectively routed to the right channels by enabling partitioning that accurately reflects the data being queried.(iii)Wireless Broadcast Channels: Each partition is assigned to a different channel to balance the load and broadcast the data.(iv)XML Query Results Aggregator: After the queries are processed, the mobile device’s XML Query Results Aggregator combines the results from the various channels or cached data to create the final query result, which is then displayed to the user.

### 3.3. Workflow of the Proposed XML Querying Scheme


(i)Partition of the XML data: The proposed method for improving XML querying over wireless broadcast channels begins by dividing the XML data into segments. The data are divided into partitions.(ii)Assign partitions to channels: Each partition is then assigned to a specific channel for broadcasting. Each partition is broadcast over a specific channel.(iii)Query routing: When a query is received, it is routed via the Query Router. Queries are routed via the Query Router.Query Decomposition: If necessary, the query is split into sub-queries and routed to the relevant partitions.Query Caching: The system also includes query caching, which checks to see if the results are already cached; if not, the query is executed, and the results are cached for later use.Query Broadcasting: Queries can be broadcast to all channels if required.(iv)Aggregate results: The results from the partitions or cache are aggregated to form the final result.(v)Return query results: The aggregated results are returned to the user.


### 3.4. Proposed Algorithm

In this section, we are proposing two, Algorithms 1 and 2, for the proposed XML querying over wireless broadcast channels.
**Algorithm 1:** An algorithm for the proposed scheme for improving XML querying over wireless broadcast channels**Inputs:** XML data stream, wireless broadcast channels, query**Outputs:** Query resultsPartition the XML data stream into several partitions based on criteria, such as document size, type, or content.Assign each partition to a different wireless broadcast channel, ensuring that the load on each channel is balanced.Broadcast the XML data partitions over their respective channels.Implement a query routing mechanism that can route the queries to the appropriate channel or combination of channels that contain the relevant XML data partition.If using query decomposition:(i)Split the query into sub-queries based on the relevant partitions.(ii)Send each sub-query to the appropriate channel that contains the relevant XML data partition.(iii)Combine the results from each channel to generate the final result.If using query caching:(i)Check if the query and its results are already cached in the mobile device.(ii)If cached, retrieve the results from the cache and return them.(iii)If not cached, send the query to the appropriate channel that contains the relevant XML data partition and cache the results for future queries.If using query broadcasting:(i)Broadcast the query to all channels.(ii)Collect the results from each channel.(iii)Combine the results to generate the final result.Return the query results.

**Algorithm 2:** An algorithm to partition an XML data stream into several partitions based on criteria**Inputs:** XML data stream**Outputs:** Partitions of XML data streamRead in the XML data stream.Choose a partitioning criterion, such as document size, document type, or document content.For each document in the XML data stream, determine the value of the chosen criterion.Create a partition for each unique value of the chosen criterion.Assign each document to the partition that corresponds to its value of the chosen criterion.Repeat steps 3–5 for all documents in the XML data stream.Output the partitions as separate XML data streams.

This algorithm divides the XML data stream into multiple partitions based on a specified criterion, such as document size, type, or content. These partitions can be used for a variety of tasks, such as improving query performance, balancing loads, and distributing data across multiple wireless broadcast channels.

### 3.5. Implementation

Both the above Algorithms 1 and 2 were implemented using Python code. All the components were created in Python code. The tests were performed on a machine running Windows 11 Professional with an Intel (R) Core i7 CPU and 8 GB of RAM. We assume that units of a fixed size are broadcasted and accessed from the XML stream. The number of buckets is then used to calculate the access time. The sample code is attached for the reference:

import xml.etree.ElementTree as ET

import os

# Read in the XML data stream

xml_file = ‘example.xml’

tree = ET.parse(xml_file)

root = tree.getroot()

# Choose a partitioning criterion, such as document size, document type, or document content

partition_criterion = ‘document_size’

# For each document in the XML data stream, determine the value of the chosen criterion

document_values = {}

for document in root.findall(‘document’):

  if partition_criterion == ‘document_size’:

    value = os.path.getsize(document.get(‘file_path’))

  elif partition_criterion == ‘document_type’:

    value = document.get(‘type’)

elif partition_criterion == ‘document_content’:

    value = document.find(‘content’).text

  else:

    raise ValueError(‘Invalid partitioning criterion’)

  # Create a partition for each unique value of the chosen criterion

  if value not in document_values:

    document_values[value] = []

  # Assign each document to the partition that corresponds to its value of the chosen criterion

  document_values[value].append(document)

# Output the partitions as separate XML data streams

for value, documents in document_values.items():

  root_partition = ET.Element(‘documents’)

  for document in documents:

    root_partition.append(document)

  partition_file = f’partition_{partition_criterion}_{value}.xml’

  ET.ElementTree(root_partition).write(partition_file)

If partitioning by document_size, each document element should have a file_path attribute, a type attribute, or a child element named content. It reads the XML data stream, determines the partitioning criterion, calculates the value of the chosen criterion for each document, creates a partition for each unique value, and assigns each document to the partition corresponding to the value of the chosen criterion. The partitions are then generated as separate XML streams.

## 4. Analysis

The access time for the proposed scheme to improve XML querying over wireless broadcast channels is determined by several factors, including the size and complexity of the XML data stream, the number of wireless broadcast channels, the query type and complexity, and network conditions. To calculate the access time for the proposed scheme, the following parameters were used:

### 4.1. Concern Parameter’s Description


(i)Partitioning the XML data stream: The time required to partition an XML data stream is determined by the partitioning criteria and the data stream’s size and complexity. In general, partitioning takes linear time, so the access time can be estimated as O (n), where n is the size of the XML data stream.(ii)Assigning partitions to wireless broadcast channels: The time required for this step varies according to the number of wireless broadcast channels and the load-balancing algorithm used. In general, load balancing can be performed in either constant or logarithmic time, so access time can be estimated as O (log n) or O (1), where n is the number of wireless broadcast channels.(iii)Broadcasting XML data partitions: The time required to broadcast the XML data partitions varies according to network bandwidth and partition size. Broadcasting is typically performed in linear time, so the access time can be estimated as O (m), where m is the total size of the XML data partitions.(iv)Query routing mechanism:*Query decomposition:* The time required for query decomposition varies according to the complexity of the query and the number of partitions involved. In general, query decomposition can take either linear or logarithmic time, so access time can be estimated as O (k log n) or O (k), where k is the number of partitions involved.*Query caching:* The time required for query caching is determined by the cache lookup time and network latency. If the query and its results are already cached, the estimated access time is O (1). If the query is not cached, the access time can be calculated as O (log n) or O (1), depending on the query routing algorithm used.*Query broadcasting:* The time required to broadcast a query is determined by the network bandwidth and the query’s size. In general, broadcasting can be performed in linear time, so the access time can be estimated as O (p), where p is the query size.(v)Combining query results: The time required to combine query results is determined by the number of partitions used and the complexity of the result aggregation algorithm. In general, result aggregation can take either linear or logarithmic time, so access time can be estimated as O (k log n) or O (k), where k is the number of partitions involved.(vi)Returning query results: The time required to return query results is determined by the network latency and the size of the result set. Returning results typically takes linear time, so the access time can be estimated as O (r), where r is the size of the result set.


It is important to take into account data partitioning, load balancing, query routing, and network conditions when determining the access time for XML querying over wireless broadcast channels. To calculate access time, we can use a general equation that accounts for these factors. Access time (*T_access_*) can be expressed as
*T_access_* = *T_partition_* + *T_transmission_* + *T_routing_* + *T_processing_*(1)
where*T_partition_* is the time required to partition the XML data stream.*T_transmission_* is the time needed to transmit the data over the wireless broadcast channels.*T_routing_* is the time taken to route the query to the appropriate channel(s).*T_processing_* is the time required to process the query and retrieve the relevant data.

These parameters are further defined as***Partitioning Time (T_partition_)***

This is the time taken to divide the XML data into partitions. It depends on the size and complexity of the XML document and can be influenced by the partitioning algorithm used.
*T_partition_* = *f* (*size_xml_*, *complexity_xml_*)(2)



**
*Transmission Time (T_transmission_)*
**



This is the time it takes to transmit the data over the wireless channels. It depends on the size of the data partition, the bandwidth of the channel, and the number of channels.
*T_transmission_* = (*Size_partition_*/*Bandwidth_channel_*) × *number_channels_*(3)



**
*Routing Time (T_routing_)*
**



This is the time required to route the query to the correct channel(s). It depends on the complexity of the routing algorithm and the number of channels.
*T_routing_* = *f* (*complexity_routing_*, *number_channels_*)(4)



**
*Processing Time (T_processing_)*
**



This is the time needed to process the query and retrieve the data. It depends on the complexity of the query and the efficiency of the query processing mechanism.
*T_processing_* = *f* (*complexity_query_*, *efficiency_processing_*)(5)

Combining these components, the formula for calculating the access time *T_access_* can be expressed as
*T_access_* = *f* (*size_xml_*, *complexity_xml_*) + [(Size_partition_/Bandwidth_channel_) × number_channels_] + *f* (*complexity_routing_*, *number_channels_*) + *f* (*complexity_query_*, *efficiency_processing_*)(6)

This formula provides a framework for estimating the access time for XML querying over wireless broadcast channels. We used the complexity of XML data as a complexity factor on a scale of 1 to 3: 3: High, 2: Medium, 1: Low.

### 4.2. Sample Data


(i)XML Data Size and Complexity:Size of XML data (*size_xml_*): 10 MBComplexity of XML data (*complexity_xml_*): Medium (2)(ii)Data Partitioning:Number of partitions: 5Time to partition each MB of XML data: 0.1 s per MBPartitioning time for each partition (*T_partition_*): *size_xml_* × 0.1 × *complexity_xml_*/*number_partitions_*(iii)Transmission:Size of each partition (*size_partition_*): 2 MB (since 10 MB/5 partitions)Bandwidth of each channel (*bandwidth_channel_*): 1 MBpsNumber of channels (*number_channels_*): 5Transmission time for each partition(iv)Routing:Complexity of routing (*complexity_routing_*): Low (1)Routing time (*T_routing_*): 0.5 s(v)Query Processing:Complexity of query (*complexity_query_*): High (3)Efficiency of processing (*efficiency_processing_*): Medium (processing time is 1 s per query complexity unit)Processing time (*T_processing_*): *complexity_query_* × *efficiency_processing_*


### 4.3. Sample Data Ranges

Further, we used the following ranges for each parameter:Size of XML data (*size_xml_*): 5 MB, 10 MB, 15 MBComplexity of XML data (*complexity_xml_*): Low (1), Medium (2), High (3)Number of partitions: 3, 5, 7Bandwidth of each channel (*bandwidth_channel_*): 1 MBps, 2 MBpsNumber of channels (*number_channels_*): 3, 5, 7Complexity of routing (*complexity_routing_*): Low (1), Medium (2), High (3)Complexity of query (*complexity_query_*): Low (1), Medium (2), High (3)Efficiency of processing (*efficiency_processing_*): Low (0.5 s per unit), Medium (1 s per unit), High (1.5 s per unit)

We assumed a fixed routing time based on the complexity level:Low: 0.5 sMedium: 1.0 sHigh: 1.5 s

Table 1 depicts the access time of the proposed indexing scheme. Each row illustrates how changing the parameters affects the total access time. This demonstrates how different XML sizes, partition numbers, bandwidths, routing complexities, and query complexities affect access time when querying XML over wireless broadcast channels.

## 5. Comparative Study

In this section, we present a detailed comparison of the proposed scheme to existing data placement schemes based on a variety of parameters. The results demonstrate the effectiveness of our proposed scheme compared to existing XML data placement schemes.

### 5.1. Comparison of Access Time of the Proposed Scheme with Various Data Placement Schemes

To provide a comparative study, the proposed scheme is compared to other XML data placement techniques. We compare the access time of these techniques under various scenarios:(i)The proposed scheme implements data partitioning, load balancing, and query routing.(ii)The broadcast disk method rotates data into predefined segments [17].(iii)Index-based methods use indexing to improve query performance [18].(iv)Sequential broadcast means broadcasting all data sequentially [19].

We consider the following sample data for comparison:Size of XML data (*size_xml_*): 10 MBComplexity of XML data (*complexity_xml_*): Medium (2)Number of partitions (for Proposed Scheme): 5Bandwidth of each channel (*bandwidth_channel_*): 1 MBpsNumber of channels (*number_channels_*): 5Routing Complexity (for Proposed Scheme): Medium (2)Query Complexity: Medium (2)Processing Efficiency: Medium (1 s per unit)

The comparison of the access time of the proposed scheme against existing XML data placement techniques is tabulated in Table 2 and Figure 3 The proposed scheme achieves a balance between access time and an efficient query routing mechanism. Because of periodic data segments, the broadcast disk method has a slightly faster access time. The index-based method is efficient because it allows for quick index lookups. The sequential broadcast has the longest access time because it waits for the entire data broadcast. This table compares the proposed scheme’s access time benefits to other existing techniques, highlighting the trade-offs. The proposed scheme strikes a balance between partitioning, transmission, routing, and processing times, giving it a competitive advantage in scenarios requiring optimal access times.

### 5.2. Comparison of Various Performance Metrics of the Proposed Scheme with Various XML Data Placement Schemes

This section compares the proposed algorithm for improving XML querying over wireless broadcast channels with existing algorithms. Several performance metrics were defined, and a comparative analysis was conducted. The performance metrics and definitions used in this analysis are as follows:*Query Response Time (QRT):* The time it takes to receive results after issuing a query.*Network Traffic (NT):* The amount of data sent across the network during the querying process. This refers to the total amount of data transmitted during querying, including any overhead.*Resource Utilization (RU):* The extent to which system resources (CPU, memory, and bandwidth) are utilized. This parameter is calculated as the percentage of CPU, memory, and bandwidth used during the query.

#### Sample Data and Assumptions

We used the following sample data for the comparison:Size of XML data (*size_xml_*): 10 MBNumber of Partitions: 5Bandwidth of each channel (*bandwidth_channel_*): 1 MBpsQuery Complexity: Medium (2 units)Processing Efficiency: Medium (1 s per unit)Routing Complexity for Proposed Scheme: Medium (1 s)

Comparative data of various performance metrics of the proposed scheme against existing XML data placement techniques is tabulated in Table 3 and depicted in Figure 4. The proposed scheme takes a balanced approach, with moderate query response time, efficient network traffic, and balanced resource utilization. The broadcast disk method provides a competitive query response time while maintaining moderate network traffic and resource utilization. The index-based method improves query response time, but at the expense of increased network traffic and resource utilization. Compared to other methods, sequential broadcast has the highest query response time and resource utilization. This quantitative comparison shows that the proposed scheme effectively balances query response time, network traffic, and resource utilization, making it a viable alternative to existing methods for XML querying via wireless broadcast channels.

### 5.3. Comparison of Access Time of the Proposed Scheme with Centralized Querying and Unpartitioned Broadcasting XML Data Placement Schemes

To provide a comparative study, the proposed scheme is compared to other XML data placement techniques. We compare the access time of these techniques under various scenarios:

The proposed scheme implements data partitioning, load balancing, and query routing.(i)Centralized Querying [23].(ii)Unpartitioned Broadcasting [24].

#### Assumptions and Sample Data


Size of XML data (*size_xml_*): 10 MBComplexity of XML data (*complexity_xml_*): Medium (2)Number of partitions (for Proposed Scheme): 5Bandwidth of each channel (*bandwidth_channel_*): 1 MBpsNumber of channels (*number_channels_*): 5Routing Complexity for Proposed Scheme: Medium (1 s)Query Complexity: Medium (2 units)Processing Efficiency: Medium (1 s per unit)Centralized Querying Complexity: Additional 2 s for centralized processing


The comparison of the access time of the proposed scheme against existing XML data placement techniques is tabulated in Table 4 and depicted in Figure 5. In terms of total access time, the proposed scheme outperforms both centralized querying and unpartitioned broadcasting methods. By partitioning the XML data and balancing the load across multiple wireless broadcast channels, the proposed scheme reduces transmission time. While efficiently routing and processing queries, this resulted in a total access time of 12 s, compared to 14 s for both centralized querying and unpartitioned broadcasting.

### 5.4. Comparison of Various Performance Metrics of the Proposed Scheme with Centralized Querying and Unpartitioned Broadcasting XML Data Placement Schemes

In this section, we compare the proposed scheme for improving XML querying over wireless broadcast channels to two other algorithms: centralized querying and unpartitioned broadcasting XML data placement schemes. Centralized querying stores XML data in a central database and sends the query to a server for processing. The server processes the query and sends the results to the mobile device. Unpartitioned broadcasting sends the entire XML data stream over the wireless channel while the query is processed on the mobile device.

We evaluated the following performance metrics:(i)**Query Response Time (QRT):** Time taken from issuing a query to receiving the results.(ii)**Network Traffic (NT):** Amount of data transmitted over the network during the query process.(iii)**Resource Utilization (RU):** Utilization of system resources (CPU, memory, bandwidth) during the query process.

The comparison of various performance metrics of the proposed scheme against centralized querying and unpartitioned broadcasting techniques is tabulated in Table 5 and depicted in Figure 6. Our proposed scheme provides the best balance between low query response time and moderate network traffic and resource utilization. Due to the need to transmit data to and from the central server, centralized querying has the highest response time and network traffic. These factors result in high resource utilization. In terms of query response time and network traffic, unpartitioned broadcasting outperforms centralized querying, but it still consumes a lot of resources and takes longer to respond than the proposed scheme.

The proposed scheme significantly reduces query response time when compared to centralized querying and unpartitioned broadcasting. The scheme reduces latency when accessing large XML documents by partitioning the data stream and distributing it across multiple wireless broadcast channels. The query routing mechanism ensures that queries are routed to the appropriate partitions, thereby improving response times.

Our proposed scheme efficiently distributes the load across multiple channels, reducing overall network traffic congestion. Centralized querying frequently results in bottlenecks because all data queries are routed through a single point. While unpartitioned broadcasting may overload the network with redundant data transmissions. The proposed scheme addresses these issues through intelligent partitioning and load balancing, resulting in more efficient network utilization.

The proposed scheme optimizes resource utilization by distributing data and query processing across multiple channels. This distribution avoids overloading any single channel while making better use of the available bandwidth. In contrast, centralized querying can put a strain on server resources, and unpartitioned broadcasting fails to capitalize on the potential of parallel processing.

This comparison highlights the efficiency of the proposed scheme in improving XML querying over wireless broadcast channels, providing faster query response times, reduced network traffic, and balanced resource utilization compared to centralized querying and unpartitioned broadcasting.

## 6. Conclusions

With the proliferation of mobile devices and wireless networks, accessing and querying large amounts of data via wireless broadcast channels has become critical. XML is a popular data format for representing and exchanging data. Querying XML data is a critical task in many applications. Wireless broadcast channels, however, may produce high latency, poor bandwidth utilization, and poor energy efficiency when querying large XML datasets. To address these issues, we proposed an approach for improving XML querying over wireless broadcast channels. The proposed scheme incorporates key parameters like data partitioning, load balancing, query routing, and result aggregation. The proposed scheme involves partitioning the XML data stream into several partitions based on criteria such as document size, type, or content. The proposed scheme assigns each partition to a separate wireless broadcast channel to balance the load on each channel. The proposed scheme also includes a query routing mechanism, which can route queries to the appropriate channel or combination of channels. These channels include the relevant XML data partition.

In our comparative study, we compared the performance of the proposed scheme for improving XML querying over wireless broadcast channels. The proposed scheme balances partitioning, transmission, routing, and processing times, giving it a competitive advantage in scenarios requiring optimal access times. The proposed scheme takes a balanced approach, with moderate query response time, efficient network traffic, and balanced resource utilization. The proposed scheme reduces transmission time by partitioning the XML data and balancing the load across multiple wireless broadcast channels. While efficiently routing and processing queries, this resulted in a total access time of 12 s, compared to 14 s for both centralized querying and unpartitioned broadcasting. The proposed scheme significantly reduces query response time when compared to centralized querying and unpartitioned broadcasting. This scheme reduces latency when accessing large XML documents by partitioning the data stream and distributing it across multiple wireless broadcast channels.

In the future, dynamic factors like network congestion or varying query loads over wireless broadcast channels will be incorporated. As a result, the system may become even more resilient and adaptable to the real world. Lower access times and more balanced resource utilization across a variety of network environments and usage scenarios would be possible for the system if it could adjust to these dynamic factors. 

## Figures and Tables

**Figure 1 sensors-24-07206-f001:**
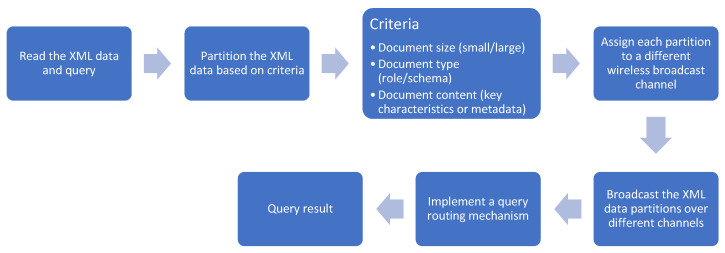
Method of proposed XML querying over wireless broadcast channels.

**Figure 2 sensors-24-07206-f002:**
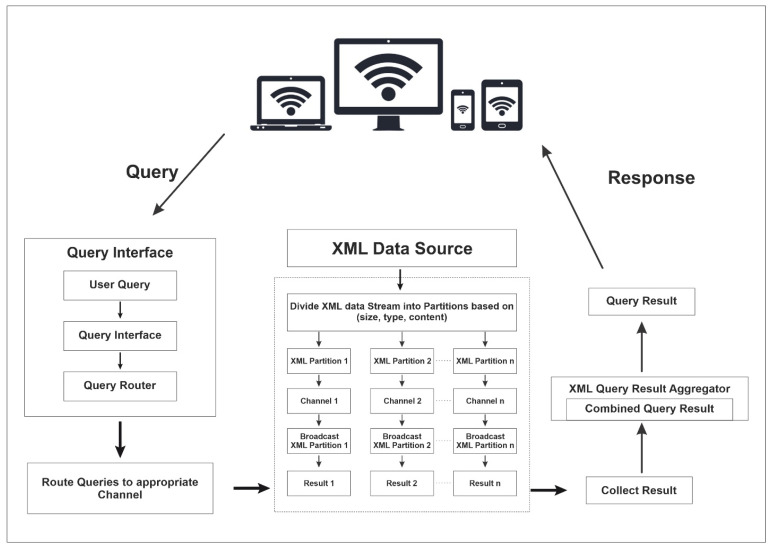
The architecture of the proposed XML querying scheme.

**Figure 3 sensors-24-07206-f003:**
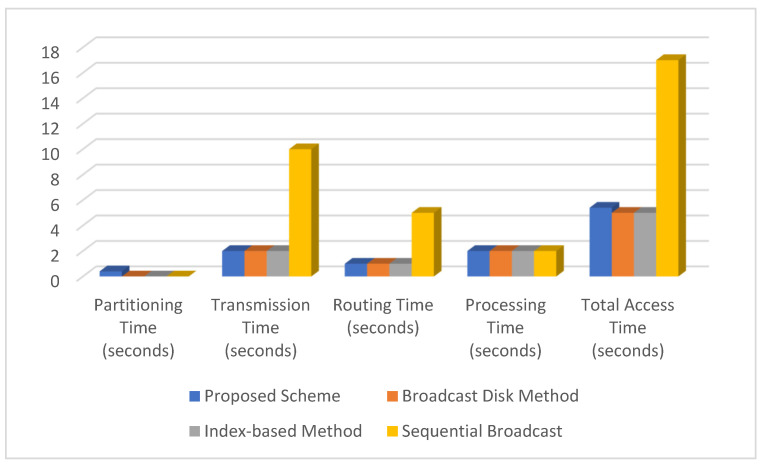
Comparison of the access time of the proposed scheme against existing XML data placement techniques.

**Figure 4 sensors-24-07206-f004:**
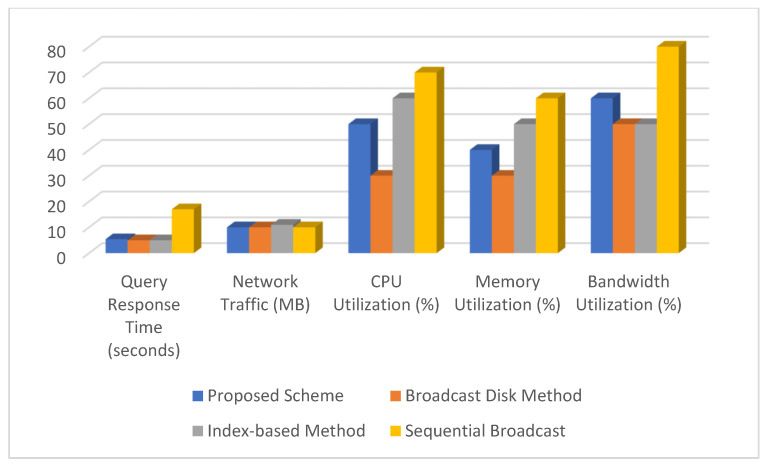
Comparison of the various performance metrics of the proposed scheme against existing XML data placement techniques.

**Figure 5 sensors-24-07206-f005:**
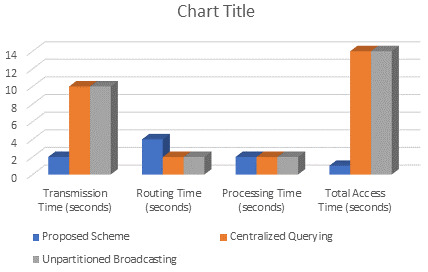
Comparison of the various performance metrics of the proposed scheme against centralized querying [23] and unpartitioned [24] broadcasting techniques.

**Figure 6 sensors-24-07206-f006:**
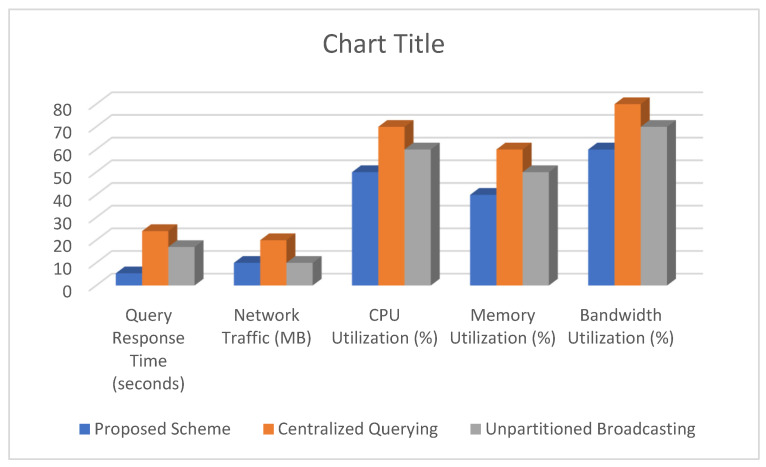
Comparison of the various performance metrics of the proposed scheme against centralized querying and unpartitioned broadcasting techniques.

**Table 1 sensors-24-07206-t001:** Access time of the proposed XML data placement scheme.

**Size of XML (MB)**	**XML Complexity**	**Partitions**	**Bandwidth (MBps)**	**Channels**	**Routing Complexity**	**Query Complexity**	**Processing Efficiency (s/unit)**	**Access Time (s)**
5	Low (1)	3	1	3	Low (1)	Low (1)	0.5	3.17
10	Medium (2)	5	1	5	Medium (2)	Medium (2)	1	6.9
15	High (3)	7	2	7	High (3)	High (3)	1.5	9.64
10	Low (1)	5	2	5	Low (1)	High (3)	0.5	5.9
5	High (3)	3	1	3	Medium (2)	Medium (2)	1	6.67
15	Medium (2)	7	1	7	Low (1)	Low (1)	1.5	7.5

**Table 2 sensors-24-07206-t002:** Comparison of the access time of the proposed scheme against existing XML data placement techniques.

	Technique	Proposed Scheme	Broadcast Disk Method [20]	Index-Based Method [21]	Sequential Broadcast [22]
Parameters					
Partitioning Time (seconds)		0.4	0	0	0
Transmission Time (seconds)		2	2	2	10
Routing Time (seconds)		1	1	1	5
Processing Time (seconds)		2	2	2	2
Total Access Time (seconds)		5.4	5	5	17

**Table 3 sensors-24-07206-t003:** Comparison of various performance metrics of the proposed scheme against existing XML data placement techniques.

Metric	Proposed Scheme	Broadcast Disk Method [20]	Index-Based Method [21]	Sequential Broadcast [22]
Query Response Time (seconds)	5.4	5	5	17
Network Traffic (MB)	10	10	11	10
CPU Utilization (%)	50	30	60	70
Memory Utilization (%)	40	30	50	60
Bandwidth Utilization (%)	60	50	50	80

**Table 4 sensors-24-07206-t004:** Comparison of the access time of the proposed scheme against existing XML data placement techniques.

	Technique	Proposed Scheme	Centralized Querying [23]	Unpartitioned Broadcasting [24]
Parameters				
Partitioning Time (seconds)		4	NA	NA
Transmission Time (seconds)		2	10	10
Routing Time (seconds)		4	2	2
Processing Time (seconds)		2	2	2
Total Access Time (seconds)		1	14	14

**Table 5 sensors-24-07206-t005:** Comparison of various performance metrics of the proposed scheme against centralized querying and unpartitioned broadcasting techniques.

Metric	Proposed Scheme	Centralized Querying [23]	Unpartitioned Broadcasting [24]
Query Response Time (seconds)	5.4	24	17
Network Traffic (MB)	10	20	10
CPU Utilization (%)	50	70	60
Memory Utilization (%)	40	60	50
Bandwidth Utilization (%)	60	80	70

## Data Availability

The data used to support the findings of this study are available from the corresponding author upon request.

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
