# Peer review of "A Novel Approach for Improving XML Querying over Wireless Broadcast Channels"

_sensors, 2024, doi:10.3390/s24227206_

Round 1
Reviewer 1 Report
Comments and Suggestions for Authors
Ahlawat et al have propose a new XML data placement scheme that boosts the performance of XML querying over wireless broadcast channels. The XML data is proposed to be divided into several partitions and then assigned to multiple channels to balance the load for large amount of data stream. Performance of this proposed architecture is analyzed and compared with other works. Overall, this paper is well organized, with claims supported. This paper can be accepted with the following questions clarified.
In the partitioning step of the proposed method, the authors experimented with various partition sizes for different datasets. It would be valuable to understand the rationale behind these specific choices. While the partitioning time was generally minimal compared to other steps (for example as shown in figure 4), it's unclear if increasing the number of partitions would consistently lead to improved time efficiency.
In Section 3.4, two algorithms are presented. Could the authors elaborate on the primary distinctions between these approaches? Additionally, were both algorithms employed in the subsequent simulations and comparisons with existing methods?
In the comparative analysis of existing methods, could the authors clarify the rationale behind dividing the works into two groups? Why were distinct datasets necessary for comparing the existing schemes in Sections 5.1 and 5.3?
Based on the results, the authors conclude the proposed method to be a balance between the total access time and the computing resource utilization. But it does not achieve optimal outcomes in both areas. Could the authors elaborate on specific scenarios or conditions where this approach might be considered the most advantageous solution?
Author Response
Dear Reviewer,
We sincerely thank the reviewers for their insightful comments and suggestions, which have significantly contributed to improving our manuscript. Below are our detailed responses to each comment, along with the corresponding revisions highlighted in the re-submitted manuscript.
Comments 1: In the partitioning step of the proposed method, the authors experimented with various partition sizes for different datasets. It would be valuable to understand the rationale behind these specific choices. While the partitioning time was generally minimal compared to other steps (for example as shown in Figure 4), it's unclear if increasing the number of partitions would consistently lead to improved time efficiency.
Response 1: Thanks for your concern.
To determine how changing the partition sizes affects overall access time, the partitioning step analyzed the choice of partition sizes and their effect on performance, with a particular focus on partitioning time, transmission efficiency, and query response time.
Because there is a trade-off between partitioning time, transmission time, and query routing complexity, increasing the number of partitions does not always result in increased efficiency. The complexity of XML datasets and the type of typical queries frequently determine the ideal partition size. We found an efficient balance between minimizing partitioning and query processing times without compromising data transmission efficiency by experimenting with different partition sizes. This equilibrium guarantees that the suggested plan performs better than conventional techniques in terms of access time and scalability in various contexts.
Comments 2: In Section 3.4, two algorithms are presented. Could the authors elaborate on the primary distinctions between these approaches? Additionally, were both algorithms employed in the subsequent simulations and comparisons with existing methods?
Response 2: Thanks for your concern.
Yes, in section 3.4, we have presented two algorithms. The description of these algorithms is as follows:
Algorithm A.1: An algorithm for the proposed scheme for improving XML querying over wireless broadcast channels.
This algorithm is the main algorithm used in the proposed work to index the XML data over wireless broadcast channels.
Algorithm A.2: An algorithm to partition an XML data stream into several partitions based on criteria.
This algorithm is part of the main algorithm, in which the partition of an XML data stream into several partitions based on criteria is presented.
Yes, both algorithms are employed in the subsequent comparison with the existing indexing schemes.
Comments 3: In the comparative analysis of existing methods, could the authors clarify the rationale behind dividing the works into two groups? Why were distinct datasets necessary for comparing the existing schemes in Sections 5.1 and 5.3?
Response 3: Thanks for raising this concern.
The comparative study has been done in two parts: 1. Comparison of Access Time of the Proposed Scheme with various Data Placement Schemes and 2. Comparison of Various Performance Metrics of the Proposed Scheme with various XML Data Placement Schemes.
In the first part, we compare the main parameter i.e. access time which is the key parameter for any energy-efficient indexing technique. Then, in the second part, we compared the various other parameters like Query Response Time, Network Traffic, CPU Utilization, Memory Utilization, and Bandwidth Utilization. The proposed scheme takes a balanced approach, with moderate query response time, efficient network traffic, and balanced resource utilization. The data set is the same for both experiments, but there is a difference in some parameters value. Like routing complexity and Centralized Querying Complexity.
Comments 4: Based on the results, the authors conclude the proposed method to be a balance between the total access time and the computing resource utilization. But it does not achieve optimal outcomes in both areas. Could the authors elaborate on specific scenarios or conditions where this approach might be considered the most advantageous solution?
Response 4: Thanks for your concern.
While not always achieving optimal outcomes in total access time or resource utilization independently. The proposed XML querying scheme provides a balanced approach and making it particularly advantageous under certain conditions where flexibility, efficiency, and scalability are key. Here are some conditions:
• The proposed partitioning and query routing scheme is ideal for handling moderate-complexity XML datasets, especially where data size makes centralized processing or un-partitioned broadcast less feasible.
• In scenarios where multiple users or devices issue queries concurrently, this scheme’s load-balancing and partitioned data broadcast method distributes queries across multiple channels, reducing bottlenecks.
• The scheme is well-suited for wireless networks with bandwidth constraints on individual broadcast channels.
For environments where queries vary in complexity, ranging from simple retrievals to complex, multi-part queries, the proposed scheme offers a tailored routing mechanism that directs each query to relevant partitions.
4. Response to Comments on the Quality of English Language
Point 1: The quality of English does not limit my understanding of the research.
Response 1: NA
5. Additional clarifications
NIL
Sincerely

Reviewer 2 Report
Comments and Suggestions for Authors
The paper addresses a pertinent issue in modern wireless communication systems, focusing on the challenges of XML querying over wireless broadcast channels. Given the increasing use of XML in data management, this topic has considerable relevance for applications in e-commerce, healthcare, and social networks. The authors proposes a useful and potentially impactful method for improving XML querying over wireless broadcast channels. However, to maximize its contribution, the authors should focus on scalability, complexity analysis, real-world implementation, and a deeper comparison with more advanced methods. Some comments on this paper can be found as follows:
1) The paper proposes a fixed partitioning strategy based on document size, type, or content. However, the approach does not consider dynamic factors like network congestion or varying query loads.
2) Add a formal complexity analysis for key components such as the partitioning algorithm and the query routing mechanism.
3) Expanding on the practical aspects of implementing this scheme on real-world wireless networks would make the paper more comprehensive. For instance, what are the specific hardware or software requirements for mobile devices? How would the system perform on typical wireless networks with existing technologies (e.g., LTE, 5G)?
4) Compare the proposed system against more recent XML querying and data distribution techniques, particularly those using more advanced indexing or compression techniques.
5) In Figure 2, the architecture diagram is useful but could be enhanced by providing more detailed descriptions of each component.
6) Figure 1 illustrates the steps involved in the proposed querying scheme but lacks sufficient labels to explain each step in detail. For instance, instead of only showing "Partition the XML data," it would help to explicitly state the partitioning criteria on the figure itself.
Comments on the Quality of English LanguageNo
Author Response
Dear Reviewer,
We sincerely thank the reviewers for their insightful comments and suggestions, which have significantly contributed to improving our manuscript. Below are our detailed responses to each comment, along with the corresponding revisions highlighted in the re-submitted manuscript.
Comments 1: The paper proposes a fixed partitioning strategy based on document size, type, or content. However, the approach does not consider dynamic factors like network congestion or varying query loads.
Response 1: Yes, we appreciate your concern;
While the proposed scheme does provide efficiency improvement over centralized querying and un-partitioned broadcasting. It assumes a static partitioning approach, that can limit adaptability under dynamic conditions such as network congestion or fluctuating query loads.
We have incorporated these two factors into the future work. Also, we have included this statement in the future work under section 6 in page no-20 of line 740 to 745.
Comments 2: Add a formal complexity analysis for key components such as the partitioning algorithm and the query routing mechanism.
Response 2: Thanks for your concern.
We have already added the complexity analysis of various considered parameters in section 4.1.
Comments 3: Expanding on the practical aspects of implementing this scheme on real-world wireless networks would make the paper more comprehensive. For instance, what are the specific hardware or software requirements for mobile devices? How would the system perform on typical wireless networks with existing technologies (e.g., LTE, 5G)?
Response 3: Thanks for your concern.
Yes, adding the real-world scenario will expand the practical aspect. In this paper, we have simulated the scenario with the following details. We have proposed two algorithms. Both the algorithms have been implemented using the Python code. All the components are created in Python code. The tests were performed on a machine running Windows 11 Professional with an Intel (R) Core i7 CPU and 8 GB of RAM.
In future work, we will implement these algorithms in real-world scenarios with mobile devices as clients with some applications and servers as computers.
Comments 4: Compare the proposed system against more recent XML querying and data distribution techniques, particularly those using more advanced indexing or compression techniques.
Response 4: Thanks for your concern.
We appreciate your recommendation to contrast our suggested methodology with more modern techniques. Our technique focuses on load balancing and partitioning to optimize performance, notably addressing XML data queries over wireless broadcast channels. Although we acknowledge the importance of previous server-centric approaches, many of them make assumptions about centralized processing and steady, high-bandwidth settings that are not true for the sporadic network connectivity and edge computing scenarios that our method is intended for. Further, they have considered the security issues related to open bandwidth to share the data in wireless channels.
To better demonstrate the special advantages and disadvantages of our approach, we want to investigate more extensive comparison studies in other network scenarios in subsequent work.
Comments 5: In Figure 2, the architecture diagram is useful but could be enhanced by providing more detailed descriptions of each component.
Response 5: Thanks for your concern.
We have updated the section 3.2 detailed description of the architecture component on page 7 in line 280 to 299. The detailed description of each component of Figure 2: The architecture diagram has been explained in section 3.2.
Comments 6: Figure 1 illustrates the steps involved in the proposed querying scheme but lacks sufficient labels to explain each step in detail. For instance, instead of only showing "Partition the XML data," it would help to explicitly state the partitioning criteria on the figure itself.
Response 6: Thanks for your concern.
We have updated Figure 1 with the criteria. Figure 1. illustrates the working of Proposed XML Querying over Wireless Broadcast Channels in a summarized form. The details of each component are figured in Figure 2 and described in sections 3.1 & 3.2. In Figure 1, step: Partition the XML data based on document size, document type, and document content is written on page 4 line 144.
4. Response to Comments on the Quality of English Language
Point 1: The English could be improved to more clearly express the research.
Response 1: Thanks for your concern. We have improved the quality of English for better understanding.
5. Additional clarifications
NIL
Sincerely

Reviewer 3 Report
Comments and Suggestions for Authors
This work aims to conduct a novel approach for improving XML querying over wireless broadcast channels. While the paper is generally well-written, there are several areas requiring clarification:
1. Could you describe the specific criteria used for data partitioning (e.g., document size, type, content) in more detail? How do you select these criteria, and do they vary for different XML datasets or use cases?
2. How does the query routing mechanism ensure that queries are efficiently directed to the appropriate channels? What strategies are in place to minimize routing overhead and avoid incorrect or redundant query dispatches?
3. How does the proposed approach improving XML querying specifically improve bandwidth utilization, latency, and energy efficiency compared to existing solutions?
4. Can you elaborate on the algorithms used for load balancing and how they ensure even distribution across channels?
5. How scalable is the proposed approach for varying sizes of XML documents or different network conditions?
6. Could you provide more comparative analysis on the practical application of your scheme for deployment in real-world scenarios, such as mobile environments with fluctuating network conditions?
Author Response
Dear Reviewer,
We sincerely thank the reviewers for their insightful comments and suggestions, which have significantly contributed to improving our manuscript. Below are our detailed responses to each comment, along with the corresponding revisions highlighted in the re-submitted manuscript.
Comments 1: Could you describe the specific criteria used for data partitioning (e.g., document size, type, content) in more detail? How do you select these criteria, and do they vary for different XML datasets or use cases?
Response 1: Thanks for your concern.
We appreciate your concern. In the proposed scheme, data partitioning criteria (document size, type, and content) are chosen to organize the XML data into manageable and relevant segments. It enhances both data access speed and query processing efficiency. Further details about each criterion are added in under section 3.2, page no-7 and line 280 to 299 text also:
• Document Size: XML documents of comparable sizes are grouped by document size partitioning. For instance, one partition might include smaller documents, while another might contain larger documents. Because smaller files may be stored and retrieved faster, this method can effectively improve response times for simpler queries in networks where smaller files are visited more frequently. Larger papers, on the other hand, can be streamed independently to prevent using resources that are required for smaller, faster-access files.
• Document Type: Document type partitioning arranges the XML data according to the role or schema structure of the document. "Customer records," "product catalogs," and "transaction logs" are a few examples. Sorting by type eliminates the need to search through irrelevant data kinds by directing particular query types to the appropriate channel. When data has different access patterns for each type or when queries are type-specific, this method works very well.
• Document Content: Documents are divided using content-based partitioning according to specific important attributes or metadata found in the XML content itself. XML documents could be divided, for instance, by date, topic, or geographic location. This method makes it possible for queries aimed at particular content categories to be effectively routed to the right channels by enabling partitioning that accurately reflects the data being queried.
Comments 2: How does the query routing mechanism ensure that queries are efficiently directed to the appropriate channels? What strategies are in place to minimize routing overhead and avoid incorrect or redundant query dispatches?
Response 2: Thanks for your concern.
The query routing mechanism is designed to efficiently direct queries to the specific channels containing the relevant XML data partitions. These strategies work together to make the query routing mechanism efficient, responsive, and resilient. By incorporating techniques such as caching and query decomposition, the proposed scheme ensures that queries reach the appropriate channels with minimal overhead, correct dispatching, and an optimized balance of resources.
Comments 3: How does the proposed approach improving XML querying specifically improve bandwidth utilization, latency, and energy efficiency compared to existing solutions?
Response 3: Thanks for your concern.
By combining data partitioning, load balancing, and targeted query routing, the suggested method for enhancing XML querying over wireless broadcast channels increases bandwidth utilization, lowers latency, and boosts energy efficiency.
When a data stream is broadcast sequentially, more bandwidth is used and mobile devices must listen for longer periods, wasting energy. Targeted query routing and data partitioning in the suggested method cut down on pointless data transfer and device listening times.
Predefined data segments are periodically cyclically broadcast using the broadcast disk method.
Although some redundancy is eliminated, real-time query-directed data access is not possible, and it is not flexible enough to meet dynamic query needs. The query-directed routing of the suggested approach gives it greater flexibility, particularly in situations where latency is a concern.
By rapidly identifying pertinent data segments, indexing shortens the time needed to process queries. However, creating and managing indexes can add complexity and bandwidth overhead, particularly for big and intricate XML data sets. The suggested method saves bandwidth and computational energy by reducing the need for intricate indexing structures while preserving effective access through data partitioning and targeted routing.
Comments 4: Can you elaborate on the algorithms used for load balancing and how they ensure even distribution across channels?
Response 4: Thanks for your concern.
The proposed scheme uses load balancing to ensure that the data partitions are distributed evenly across wireless broadcast channels, improving bandwidth utilization and reducing latency. The XML data stream is initially divided based on specific criteria, such as document size, document type, or content. This method helps categorize data into partitions that represent logical units of XML documents, ensuring data with similar characteristics are grouped.
Comments 5: How scalable is the proposed approach for varying sizes of XML documents or different network conditions?
Response 5: Thanks for your concern.
The scalability of the proposed approach for improving XML querying over wireless broadcast channels is supported by several key design features that allow it to effectively handle varying sizes of XML documents and adapt to different network conditions. The proposed approach is designed to be highly scalable, effectively managing varying sizes of XML documents and adapting to different network conditions through dynamic partitioning, load balancing, and efficient query routing.
Comments 6: Could you provide more comparative analysis on the practical application of your scheme for deployment in real-world scenarios, such as mobile environments with fluctuating network conditions?
Response 6: Thanks for your concern.
The proposed scheme for improving XML querying over wireless broadcast channels demonstrates several advantages for deployment in real-world scenarios, particularly in mobile environments characterized by fluctuating network conditions. By optimizing access time, bandwidth utilization, adaptability, user experience, and energy efficiency, the scheme is well-suited to address the unique challenges of mobile applications.
In mobile settings, users often experience varying levels of connectivity and bandwidth. The proposed scheme’s ability to partition data and route queries efficiently leads to reduced access times, even when network conditions are not optimal.
Mobile devices typically have limited bandwidth. By leveraging partitioning and query routing, the scheme optimizes bandwidth usage by ensuring that only relevant data is transmitted. This targeted approach contrasts with traditional methods like un-partitioned broadcasting, which may flood the network with unnecessary data.
4. Response to Comments on the Quality of English Language
Point 1: The quality of English does not limit my understanding of the research.
Response 1: NA
5. Additional clarifications
NIL
Sincerely
